# Phage Therapy as a Rescue Treatment for Recurrent *Pseudomonas aeruginosa* Bentall Infection

**DOI:** 10.3390/v17010123

**Published:** 2025-01-17

**Authors:** Victor Eiferman, Pierre-Adrien Vion, Alexandre Bleibtreu

**Affiliations:** 1Service des Maladies Infectieuses et Tropicales, Hôpital Pitié Salpêtrière, APHP Sorbonne Université, 75013 Paris, France; alexandre.bleibtreu@aphp.fr; 2Service de Médecine Nucléaire, Hôpital Pitié Salpêtrière, Assistance Publique des Hôpitaux de Paris (APHP), 75013 Paris, France; pierre-adrien.vion@aphp.fr

**Keywords:** phage therapy, phagogram, Bentall procedure, endocarditis, vascular graft

## Abstract

Phage therapy is experiencing renewed interest, particularly for antibiotic-resistant infections, and may also be useful for difficult-to-treat cases where surgery to remove foreign infected material is deemed too risky. We report a case of recurrent *Pseudomonas aeruginosa* endocarditis with Bentall infection treated successfully with a combination of antibiotics and phages.

## 1. Introduction

A century after its conceptualization by d’Hérelle and 50 years after it was forgotten in France, phage therapy is back in the spotlight [1]. Phage therapy is most often considered in cases of multi-drug resistant infections, where it provides a therapeutic alternative. Beside the spread of antibiotic resistance, difficult-to-treat bacterial infection (DTI) is another area of interest for phage therapy [2]. These DTIs are caused by antibiotic-sensitive bacteria but associated with foreign material or responsible for infections that require a surgical procedure that is too risky for vulnerable patients. Phages are known for their ability to disrupt biofilms and enhance the efficacy of antibiotics through synergistic and complementary actions [3,4]. Vascular device infections are complex infections that often require medical and surgical management. Infection following the Bentall procedure, in which the aortic valve, aortic root and ascending aorta are replaced with a composite graft, is particularly challenging and associated with high mortality [5,6]. Here, we report an emblematic case of recurrent *Pseudomonas aeruginosa* endocarditis with Bentall infection treated with a combination of antibiotics and phages.

## 2. Case Presentation

A 58-year-old man with a history of aortic regurgitation and dilatation underwent aortic valve repair, aortic root replacement, and ascending aortic replacement with coronary artery reimplantation in May 2022. In late October 2022, the patient presented with neurological deterioration in a septic context. Investigations revealed bacterial prostatitis caused by multi-susceptible *P. aeruginosa*. Transesophageal echocardiography showed a perivalvular abscess, and computed tomography scan showed a 25 × 17 mm long axis collection on the anterior surface of the aortic valve prosthesis, along with a latero-ventricular pericardial effusion. Cerebral magnetic resonance imaging showed multiple emboli with disseminated small ischemic strokes. The patient was treated with a Bentall bioprosthetic valve replacement and high-dose ceftazidime for 6 weeks.

In March 2023, he presented to the hospital with fever and oxygen dependency. Multiple blood cultures were, once again, positive for *P. aeruginosa* with the same resistance profile. Transesophageal echocardiography revealed two aortic vegetations, and a cardiac scan identified an 8 mm circumferential hypodense collection, which enhanced after injection, around the prosthetic ascending aorta.

In a multidisciplinary meeting, due to the high risk of surgery, it was decided to use a medical-only strategy. A combination of ceftazidime and ciprofloxacin for three months, followed by suppressive treatment with ciprofloxacin was decided. Evaluation after six weeks of combination therapy showed persistent apyrexia and normalization of C-reactive protein (CRP) levels on the one hand, but a sternotomy fistula from the previous scar on the other hand. A computed tomography scan confirmed the fistula following a scar to retrosternal collection trajectory with a 25 × 10 × 6 cm collection surrounding the prosthetic aorta. This was drained surgically on 28 April 2023, with a simple wash and no equipment change. Cultures were sterile, as the surgery was performed under antibiotic therapy. Follow-up transesophageal echocardiography showed persistent vegetation at the antero-left cusp with no change in size (10 mm × 10 mm), and a computed tomography scan showed stability of the collection around the ascending aorta and sternal disunion with a retrosternal collection of 13 mm.

Given the stability of the lesion and the lack of clinical improvement, phage therapy targeting *P. aeruginosa* was initiated under compassionate use on 22 May 2023 for seven days. The phage susceptibility of the clinical isolate was determined in the Phaxiam-Pharma laboratory. Four phages, two myoviruses (PP 1450 and PP 1777) and two podoviruses (PP 1792, PP 1797) were tested against *P. aeruginosa*, as previously described [7]. Susceptibility was determined by the combined result of two assays. The first one was the plaque assay, which determined the ability of the test phage to induce bacterial lysis in a solid medium compared to a susceptible reference phage, thus determining the efficiency of plating (EOP). An EOP of 0 indicates that no lysis zone is observed. An EOP greater than 0.1 represents a highly effective phage, while an EOP between 0.005 and 0.1 indicates a moderately effective phage and between 0 and 0.005 a low efficiency phage [8]. The second test was the broth microdilution assay, which identified the minimum inhibitory concentration (MIC) expressed in plaque-forming units (PFU) per milliliter of phage to inhibit 80 +/− 8% of bacterial growth. The EOP values of PP 1450, PP 1777, PP1792 and 1797 were 0.606, 0.932, 0.053 and 0.206, respectively. The MIC of PP 1450, PP 1777, PP1792 and 1797 were 1.3 × 10^3^ PFU/mL, 2.0 × 10^3^ PFU/mL, >1 × 10^9^ PFU/mL and >1 × 10^9^ PFU/mL, respectively. Based on this result, myoviruses (PP 1450 and PP 1777) were selected for patient treatment, while podoviruses (PP 1792 and PP 1797) were not considered. The treatment consisted of a phage solution containing 1 mL of 10^10^ PFU of each phage diluted in 8 mL of 0.9% NaCl sterile solution and administered intravenously every 12 h for seven days. When the phages were administered, a new curative antibiotic regimen of ceftazidime and ciprofloxacin was initiated for three months, followed by maintenance therapy with ciprofloxacin monotherapy for three months.

A positive emission tomography (PET) scan performed after 3 days of phage and antibiotic treatment showed circumferential hypermetabolism around the infected aortic tube (Figure 1B,C). Subsequent PET scans at the end of bi-antibiotic therapy (September 2023) and at the end of ciprofloxacin maintenance therapy (February 2024) showed regression of the hypermetabolism. The metabolism of the Bentall walls regressed almost completely, with regression of the linear hypermetabolism resembling a fistula between the Bentall and the lower part of the sternotomy scar. In addition, the hypermetabolic focus at the lower part of the sternotomy disappeared. Clinically, the patient’s general condition improved without a fever episode, and the sternal fistula disappeared. Transesophageal echocardiography revealed no vegetation. Biologically, subsequent blood cultures were all negative, and CRP levels remained negative 12 months after phage therapy. The patient was considered cured, and antibiotics were discontinued with no relapse to date.

## 3. Discussion

This case report describes the successful use of a combination phage therapy in treating recurrent *P. aeruginosa* endocarditis following a Bentall procedure, where conventional antibiotic treatments had failed, and surgical options were too risky.

Recurrent infections associated with valve and vascular prostheses pose a significant problem due to antibiotic spread and biofilm formation, requiring, theoretically, a combination of surgical replacement and antibiotic treatment. Bentall surgical replacement is a high-risk procedure associated with high mortality, so some teams prefer to leave the implant in place with suppressive antibiotics in unsuitable patients, with encouraging results [9]. However, conservative management is associated with recurrent sepsis, which can occur in up to 50% of patients in this cohort for all vascular grafts [10], and high mortality, with up to 40% mortality for thoracic grafts [11]. FDG-PET is a valuable tool for diagnosing or excluding vascular graft infection with excellent negative and positive predictive values [12]. FDG-PET has been incorporated into the Duke criteria for prosthetic valve endocarditis [13]. Its role in monitoring infection and recovery is more controversial, but is of interest when combined with clinical status and inflammatory markers such as CRP to guide the discontinuation of antibiotic therapy [14,15].

The existing literature on phage therapy for vascular infections is limited but demonstrates considerable promise [16]. Despite in vitro studies showing synergistic effect between phages and antibiotics, real-life use remains scarce [17,18]. Five case reports have documented clinical outcomes in vascular prosthesis infections caused by *P. aeruginosa*, with inconclusive results [17,18,19,20,21]. In three of these reports, the authors reported a relative success characterized by bacterial clearance without the recurrence of bacteremia. However, two of the patients died within six months, and the third had to undergo partial replacement of the infected prosthesis [20,22,23]. The remaining two reports documented recurrences of *P. aeruginosa* bacteremia. One of the cases documented a relapse to a clonally related *Pseudomonas* infection that exhibited changes in genetic and phenotypic traits, resulting in increased antibiotic sensitivity but enhanced biofilm formation and resistance to the phages used [21]. Additionally, some teams have utilized phages to treat *P. aeruginosa* infections in left ventricular assist devices with variable outcomes [24,25]. In these reports, phage therapy was used as a last resort treatment with advanced local infection and deteriorated general health, which can complicate the assessment of its effectiveness.

The cornerstone for phage therapy use is susceptibility testing, also called phagogram. We need to test patient’s bacterial strain and use effective phages with standardized methodologies [26]. Uncertainties remain regarding the optimal phage dosage, treatment schedule, and administration methods for phage therapy [27]. These gaps in knowledge can significantly affect outcomes and explain the failures already published.

While it is conceivable that the described patient might have improved without phage therapy, the worsening clinical condition under dual antibiotic therapy, highlighted by the emergence of a new fluid collection and fistulization, strongly suggests a pivotal role for phages in his recovery. Furthermore, the only modification in therapeutic management between the second and third episodes of infection was the addition of phage therapy to the antibiotic treatment.

While case reports of phage therapy are uncommon in conditions such as endocarditis or vascular prosthesis infections, they can be a valuable resource when surgical intervention is not feasible. The patient remains relapse-free for 12 months with no evidence of recurrence to date. It is of the utmost importance to document both positive and negative outcomes from such treatments to gain a deeper understanding in phage therapy. The characterization and optimization of the pharmacokinetics of phages, as well as their combination with each other and with antibiotics, require further investigation. As several case reports and case series have shown promising results, often as a therapy of last resort, large multicenter clinical trials with standardized methodologies should be initiated to prove the efficacy of phage therapy.

## Figures and Tables

**Figure 1 viruses-17-00123-f001:**
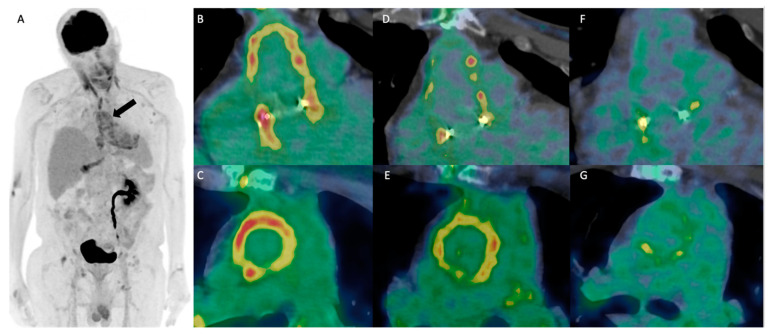
Initial FDG-PET 3D MIP (**A**) showing the circumferential hypermetabolism around the infected Bentall (indicated by arrow). Coronal and axial reoriented fused FDG-PET-CT views of the Bentall before (**B**,**C**) the introduction of phage therapy, showing heterogeneous, circumferential hypermetabolism compatible with an active infection. This anomaly gradually regressed on successive FDG-PET at 3 months (**D**,**E**) and 6 months (**F**,**G**) after phage-therapy introduction, indicating good therapeutic response. FDG-PET 3D MIP—fluorodeoxyglucose positron emission tomography three-dimensional maximum intensity projection.

## Data Availability

Data are unavailable due to privacy and ethical restrictions.

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
