# Peer review of "Phage Therapy as a Rescue Treatment for Recurrent Pseudomonas aeruginosa Bentall Infection"

_viruses, 2025, doi:10.3390/v17010123_

Round 1
Reviewer 1 Report
Comments and Suggestions for Authors
General comment
I read very carefully the manuscript by Eiferman et al. titled “Phage Therapy as a Rescue Treatment for Recurrent Pseudomonas aeruginosa Bentall Infection”. The topic is very interesting, and the successful cases of using phage therapy in complex indications such as the one presented help to validate a potential indication of this treatment in our therapeutic arsenal. The case report is clear and well presented. The proposed figure is relevant. Only the English translation could be improved.
Specific comments
Abstract
/
Introduction
- Page 1. Line 18. The spelling of the name should be changed “D'hérelle” for “d’Hérelle”
- Page 1. Lines 24 to 26. “Phages are known for their ability to disrupt biofilms and enhance the efficacy of antibiotics through synergistic and complementary actions.” Could the authors provide a reference?
Case presentation
- Page 3. Line 92. “FDG-PET, fluorodeoxyglucose positron emission tomography”. Could the authors explain what this sentence means?
Figure
/
Discussion
- Page 3. Line 95. “Pseudomonas aeruginosa” should be changed for “P. aeruginosa”
Conclusion
/
Comments on the Quality of English LanguageThe English translation could be improved.
Author Response
Comments 1 : Page 1. Line 18. The spelling of the name should be changed “D'hérelle” for “d’Hérelle”
Response 1 : Thank you for pointing this out, we have revised our manuscript. Page 1 Line 18
Comments 2: Page 1. Lines 24 to 26. “Phages are known for their ability to disrupt biofilms and enhance the efficacy of antibiotics through synergistic and complementary actions.” Could the authors provide a reference?
Response 2: We fully agree that this sentence needs appropriate references. We have added 2 references on page 1 line 26.
Harper, D.; Parracho, H.; Walker, J.; Sharp, R.; Hughes, G.; Werthén, M.; Lehman, S.; Morales, S. Bacteriophages and Biofilms. Antibiotics 2014, 3, 270–284, doi:10.3390/antibiotics3030270.
Holger, D.J.; El Ghali, A.; Bhutani, N.; Lev, K.L.; Stamper, K.; Kebriaei, R.; Kunz Coyne, A.J.; Morrisette, T.; Shah, R.; Alexander, J.; et al. Phage-Antibiotic Combinations against Multidrug-Resistant Pseudomonas Aeruginosa in in Vitro Static and Dynamic Biofilm Models. Antimicrob Agents Chemother 2023
Comment 3 : Page 3. Line 92. “FDG-PET, fluorodeoxyglucose positron emission tomography”. Could the authors explain what this sentence means?
Response 3: Thank you for your comment. We have changed the layout to make it easier to understand by adding an asterisk. It was not a sentence but an explanation of the abbreviation. Page 3 line 106-107 *FDG-PET 3D MIP: fluorodeoxyglucose positron emission tomography three-dimensional maximum intensity projection
Comment 4 : Page 3. Line 98. “Pseudomonas aeruginosa” should be changed for “P. aeruginosa”
Response 4 : Page 3. Line 110. Thank you for noticed, we are agree and have made the appropriate change.
Reviewer 2 Report
Comments and Suggestions for Authors
The study provides the potential value of phage therapy in treating infections that are difficult to address with traditional antibiotics, which is a very interesting and clinically significant field. Here are some reviews:
1. The study demonstrates the potential of phage therapy in treating recurrent Pseudomonas aeruginosa endocarditis, especially when conventional antibiotic treatments have failed and surgical risks are too high. This is an important clinical issue, and your research offers a new perspective for addressing it.
2. The article provides a detailed description of phage selection, susceptibility testing, and the treatment process. However, it is recommended that the authors provide more detailed information on phage preparation and dose selection to allow other researchers to replicate these experiments.
3. The article includes PET scan images that vividly demonstrate the effects of phage therapy. It is suggested that the discussion section further analyzes these images to emphasize the clinical effectiveness of phage therapy.
4. The authors have mentioned the potential advantages and challenges of phage therapy in the discussion, but it could be expanded to include a more comprehensive review of existing literature, particularly in comparison with other studies on phage therapy for vascular infections.
5. The conclusion section of the article clearly summarizes the main findings of the study. It is recommended that the authors briefly discuss future research directions for phage therapy and how to overcome the challenges currently faced in treatment.
6. Line 86: "Figure 1. Initial FDG-PET 3D MIP (A) showing the circumferential hypermetabolism around the infected Bentall (arrow)." The reference "(arrow)" likely pertains to an arrow in the figure caption; it is suggested to be revised to "(indicated by arrow)" to avoid ambiguity.
7. Line 94: "This case report describes the successful use of a dual-phage therapy in treating recurrent Pseudomonas aeruginosa endocarditis following a Bentall procedure, where conventional antibiotic treatments had failed, and surgical options were too risky." The term "dual-phage therapy" could be more clearly expressed as "combination phage therapy."
Overall, this research work is a valuable contribution to the field of phage therapy. After making the appropriate revisions to the comments above, I recommend accepting this article for publication.
Author Response
Comment 1: The study demonstrates the potential of phage therapy in treating recurrent Pseudomonas aeruginosa endocarditis, especially when conventional antibiotic treatments have failed and surgical risks are too high. This is an important clinical issue, and your research offers a new perspective for addressing it.
Response 1 : Thank you for your comment.
Comment 2: The article provides a detailed description of phage selection, susceptibility testing, and the treatment process. However, it is recommended that the authors provide more detailed information on phage preparation and dose selection to allow other researchers to replicate these experiments.
Réponse 2 : Thank for your comment. The phage preparation have been extensively described in a previous case report in Viruses. We added the reference to make it more clear and added a more comprehensive paragraph to illustrate phage selection.
Page 2 Line 66-78
Susceptibility was determined by the combined result of two assays. First, the plaque assay, which determines the ability of the test phage to induce bacterial lysis in a solid medium compared to a susceptible reference phage, thus determining the efficiency of plating (EOP). An EOP of 0 indicates that no lysis zone is observed. An EOP greater than 0.1 represents a highly effective phage, while an EOP between 0.005 and 0.1 indicates a moderately effective phage and between 0 and 0.005 a low efficiency phage [8]. The second test is the broth microdilution assay and identifies the minimum inhibitory concentration (MIC) expressed in plaque-forming units (PFU) per milliliter of phage to inhibit 80% +/- 8% of bacterial growth. The EOP values of PP 1450, PP 1777, PP1792 and 1797 were 0.606, 0.932, 0.053 and 0.206, respectively. The MIC of PP 1450, PP 1777, PP1792 and 1797 were 1.3x10^3 PFU/mL, 2.0x10^3 PFU/mL, >1x10^9 PFU/mL and >1x10^9 PFU/mL, respectively. Based on this result, myoviruses (PP 1450 and PP 1777) were selected for patient treatment, while podoviruses (PP 1792 and PP 1797) were not considered
Comment 3: The article includes PET scan images that vividly demonstrate the effects of phage therapy. It is suggested that the discussion section further analyzes these images to emphasize the clinical effectiveness of phage therapy.
Response 3 : Thank you for your pertinent comment. We have added a paragraph with references to clarify this point. Page 3 Ligne 119-124
FDG-PET is a valuable tool for diagnosing or excluding vascular graft infection with excellent negative and positive predictive values [12]. FDG-PET has been incorporated into the Duke criteria for prosthetic valve endocarditis [13]. Its role in monitoring infection and recovery is more controversial, but is of interest when combined with clinical status and inflammatory markers such as CRP to guide discontinuation of antibiotic therapy [14,15]
Comment 4 : The authors have mentioned the potential advantages and challenges of phage therapy in the discussion, but it could be expanded to include a more comprehensive review of existing literature, particularly in comparison with other studies on phage therapy for vascular infections.
Response 4 : A review of the literature found few reported cases of vascular infection. We provide a detail of the result of this different case report dans le paragraphe page 3-4 line 125-139
Comment 5 : The conclusion section of the article clearly summarizes the main findings of the study. It is recommended that the authors briefly discuss future research directions for phage therapy and how to overcome the challenges currently faced in treatment.
Response 5 : Thank you for your comment. We added a sentence to the conclusion to outline the challenge to phage therapy. Page 4, line 155-160. "The characterization and optimization of the pharmacokinetics of phages, as well as their combination with each other and with antibiotics, require further investigation. As several case reports and case series have shown promising results, often as a therapy of last resort, large multicenter clinical trials with standardized methodologies should be initiated to prove the efficacy of phage therapy."
Comment 6. Line 86: "Figure 1. Initial FDG-PET 3D MIP (A) showing the circumferential hypermetabolism around the infected Bentall (arrow)." The reference "(arrow)" likely pertains to an arrow in the figure caption; it is suggested to be revised to "(indicated by arrow)" to avoid ambiguity.
Response 6: Thank you for your notice. We have changed the sentence as suggested Page 3/6 Line 100 "indicated by arrow"
Comment 7. Line 94: "This case report describes the successful use of a dual-phage therapy in treating recurrent Pseudomonas aeruginosa endocarditis following a Bentall procedure, where conventional antibiotic treatments had failed, and surgical options were too risky." The term "dual-phage therapy" could be more clearly expressed as "combination phage therapy."
Response 7. Thank you for your comment, you are right it is more clear with "combination phage therapy", we have modified the sentence as suggest. Page 3 Line 109
Reviewer 3 Report
Comments and Suggestions for Authors
The manuscript presents an interesting and timely case report on the successful application of phage therapy for recurrent Pseudomonas aeruginosa endocarditis in a high-risk patient. The use of phages as an adjunct to antibiotics in vascular infections remains an underexplored area of clinical medicine, and this case adds valuable evidence to the limited literature. The article is well-structured, with clear sections that logically guide the reader through the clinical case, methodology, and outcomes. However, I have several comments that primarily concern the significant revision of the Introduction section.
Comments:
Lines 18-19. The first sentence presents information that is unlikely to be familiar to a broad audience. If this information is important, please provide references to sources for readers who may wish to gain a deeper understanding of the topic.
Lines 26-27. I also believe that the authors need to explain, in the Introduction, for the readers of the journal Viruses, what Bentall infection is and how it is related to the Bentall procedure. This information should be accompanied by references to relevant literature for a more detailed understanding of the Bentall procedure and its potential infectious complications.
Additionally, the Introduction should clarify why the authors chose two myoviruses (PP 1450 and PP 1777) and two podoviruses (PP 1792, PP 1797) in combination with antibiotics for treatment. Are there documented cases of clinical use of these phages? If so, please provide references to sources.
Line 112. “LVAD” ─ The abbreviation needs to be expanded.
Author Response
Comment 1 : Lines 18-19. The first sentence presents information that is unlikely to be familiar to a broad audience. If this information is important, please provide references to sources for readers who may wish to gain a deeper understanding of the topic.
Response 1 : Thank you for your comment, we add a reference page 1 line 19
Chanishvili, N. Phage Therapy—History from Twort and d’Herelle Through Soviet Experience to Current Approaches. In Advances in Virus Research; Elsevier, 2012; Vol. 83, pp. 3–40 ISBN 978-0-12-394438-2.
Comment 2 : Lines 26-27. I also believe that the authors need to explain, in the Introduction, for the readers of the journal Viruses, what Bentall infection is and how it is related to the Bentall procedure. This information should be accompanied by references to relevant literature for a more detailed understanding of the Bentall procedure and its potential infectious complications.
Response 2 : Thank you for this pertinent comment. We added explanation about procedure and references. Page 1 Line 26-29 "Vascular device infections are complex infections that often require medical and surgical management. Infection following the Bentall procedure, in which the aortic valve, aortic root and ascending aorta are replaced with a composite graft, is particularly challenging and associated with high mortality [5,6]"
Comment 3 : Additionally, the Introduction should clarify why the authors chose two myoviruses (PP 1450 and PP 1777) and two podoviruses (PP 1792, PP 1797) in combination with antibiotics for treatment. Are there documented cases of clinical use of these phages? If so, please provide references to sources.
Response 3 : Thank you for your comment. These phages are systematically tested against Pseudomonas aeruginosa strains by the Phaxiam laboratory. We have added a reference with the previous publication and a more detail paragraph.
Page 2 Line 63-65 Four phages, two myoviruses (PP 1450 and PP 1777) and two podoviruses (PP 1792, PP 1797) were tested against P. aeruginosa as previously described [7] Reference : Teney, C.; Poupelin, J.-C.; Briot, T.; Le Bouar, M.; Fevre, C.; Brosset, S.; Martin, O.; Valour, F.; Roussel-Gaillard, T.; Leboucher, G.; et al. Phage Therapy in a Burn Patient Colonized with Extensively Drug-Resistant Pseudomonas Aeruginosa Responsible for Relapsing Ventilator-Associated Pneumonia and Bacteriemia. Viruses 2024
Comment 4: Line 112. “LVAD” ─ The abbreviation needs to be expanded.
Thank you for your comment, we adjust the text - Page 4 Line 137 LVAD was replaced by left ventricular assist devices
Round 2
Reviewer 3 Report
Comments and Suggestions for Authors
I have reviewed the revised version of the manuscript, which includes significant improvements by the authors. Since the authors have addressed the main concerns, I can recommend this version of the manuscript for publication after minor revisions.
Minor comments:
Line 126. “...real life use remain scarce [17,18]” – "Remain" should be "remains".
Lines 127-129. “with inconclusive results (3–7) – The references “(3–7)” from the older version of the manuscript should be placed in square brackets and updated. In the revised manuscript, they should correspond to [17–21]. Please check.
Line 137. “with carriable outcomes” – Perhaps the intended term is "variable" or "mixed outcomes." Please verify.
Author Response
Comments 1 Line 126. “...real life use remain scarce [17,18]” – "Remain" should be "remains".
Response 1 Line 126 Thank you for your notice, we have changed "remain" to "remains".
Comment 2 : Lines 127-129. “with inconclusive results (3–7) – The references “(3–7)” from the older version of the manuscript should be placed in square brackets and updated. In the revised manuscript, they should correspond to [17–21]. Please check.
Response 2 Line 127-129 Thank you for your careful observation, we have corrected the reference to [17–21]
Comment 3 Line 137. “with carriable outcomes” – Perhaps the intended term is "variable" or "mixed outcomes." Please verify.
Response 3 Line 137. Thank you for your comment, we have changed carriable by variable as suggested